# Socially-Attentive Policy Optimization in Multi-Agent Self-Driving System

**Zipeng Dai**[1,*], **Tianze Zhou**[1,*], **Kun Shao**[2†], **David Henry Mguni**[2], **Bin Wang**[2], **Jianye Hao**[23]

[1]Beijing Institute of Technology, [2]Huawei Noah's Ark Lab, [3]Tianjin University

**Abstract:** As increasing numbers of autonomous vehicles (AVs) are being deployed, it is important to construct a multi-agent self-driving (MASD) system for navigating traffic flows of AVs. To prevent congestion or collision, especially in scenarios like intersection or lane merging, an AV should not only pursue the individual goal but also interact with other traffic participants. Inspired by the mixed motives like human drivers, we mainly focus on how to figure out the current interacting vehicles in MASD systems, and then model the coordination or conflicts of their goals, namely their social preferences (SPs). To this end, we propose a novel reinforcement learning method for each AV in decentralized control, called Socially-Attentive Policy Optimization (SAPO), which incorporates: (a) a self-attention module to select the most interactive traffic participant, and (b) a social-aware integration mechanism to integrate objectives of the ego AV and its interacting partner, by updating their SPs from the current observations. SAPO solves the decision-making problem of how to improve the safety and efficiency of MASD systems, without utilizing any global information for centralized control. Simulation experiments show that SAPO can successfully capture and utilize the variation of the SPs of AVs to achieve superior performance.

**Keywords:** self-driving, social preference, multi-agent, reinforcement learning

## 1 Introduction

With the development of self-driving technologies, increasing numbers of autonomous vehicles (AVs) are being deployed on public roads, and sharing the space with other traffic participants. Different from the single-agent platforms [1] built for realism in rendering a single AV under different road and weather conditions, the Multi-Agent Self-Driving (MASD) system [2, 3, 4] is designed to navigate traffic flows of multiple AVs, where each AV needs to not only pursue the individual goal but also interact with other traffic participants in vehicle interaction events (e.g., intersection and lane merging). As with human drivers and pedestrians who usually show respect to others, it is important for MASD systems to model the coordination or conflicts of the AV's goals, namely the social preferences (SPs). Fig. 1 shows the illustrative examples where interacting AVs behave differently in different SPs. Failing to incorporate an awareness of other agents' SPs can lead to increased congestion as well as more collisions between vehicles.

Enabling SPs in MASD systems produces a series of challenges: First, for the efficiency and accuracy of modeling SPs, it is necessary to determine which ranges or groups of AVs are actually interacting. For example, as shown in Fig. 1, interaction events occur in the red zone when AVs are crossing the intersection together. Second, as time goes by, the SP of an AV usually varies because it may play various roles when interacting with other AVs in different kinds or driving styles.

To this end, we aim to capture the most likely object which each AV interacts with, and then extract their SPs to improve the efficiency of MASD control algorithms. Existing motion planning models [5, 6] and safe control methods [7, 8] only focus on the physical states and subjective observations of the ego car, but fail to incorporate SPs for decision-making purposes. Multi-agent

---

* Work done as an intern at Huawei. † Corresponding author: shaokun2@huawei.com.

6th Conference on Robot Learning (CoRL 2022), Auckland, New Zealand.

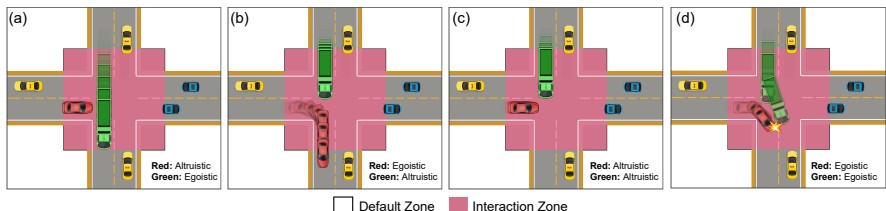

Figure 1: Examples of an MASD system, where 2 AVs are in different social preferences.

reinforcement learning (MARL) is becoming increasingly popular for several sequential decision-making scenarios [9, 10] with multiple controllable agents. Exemplar methods like QMIX [11] and MADDPG [9] are widely used in MASD systems with multiple controllable agents in fully cooperative or fully competitive settings, by defining a shared team reward or individual payoffs as the objectives. However, the SPs of these agents are fixed, which are far from the real-world settings.

**Contributions.** Our main contribution is a novel MARL method called Socially-Attentive Policy Optimization (SAPO) for independent AV control, which solves the problem of how to improve the safety and efficiency of an MASD system with the dynamic modeling of SPs. This method consists of two key components, (i) a self-attention module to select the most interactive participant for each AV, which extracts the critical entities in vehicle interaction events, (ii) a social-aware integration mechanism to integrate objectives of the ego AV and its interacting partner, by updating their SPs from the current observations. Based on a simulation platform called SMARTS [4], we validate SAPO in several MASD scenarios, and then we showcase how SAPO navigates each AV to select the interacting object and estimate the SPs in real time, which current MARL methods cannot handle.

## 2 Related Work

### 2.1 Deep Reinforcement Learning in MASD Systems

In recent research literature, MARL is widely used to navigate AVs in MASD systems. Wu *et al.* in [12] proposed an MARL-based velocity maximization framework by effectively leveraging the structure of the human driving behavior to form an efficient vehicle spacing for an intersection network. Isele *et al.* in [13] designed several exploratory actions for MARL to optimize the acceleration profile while safely avoiding collisions. Yang *et al.* in [14] proposed a credit-assignment method for interactions between actions and goals of different AVs. Peng *et al.* in [15] considered the fact that each individual agent is mostly affected by its nearby agents, and facilitate the coordination of AVs at both local and global levels in MASD systems. However, most of the existing methods only focus on how to achieve safe responses to sudden actions of the surrounding entites, or just assume that each AV has a fixed SP in an episode [15], which are limited to apply in general cases.

### 2.2 Social Preference in Vehicle Interaction

SP is increasingly discussed to better formulate interaction between AVs in self-driving scenarios. The authors in [16, 17] proposed probabilistic frameworks that efficiently learned categorical latent variables to jointly model the multi-step future motions of agents in a scene. Sekhon *et al.* in [18] proposed a novel spatial attention mechanism to encode the influence of spatially close neighbors in a manner. Apart from these powerful works about trajectory prediction, several decision-making works also start to consider how to utilize SP for exploring better control policies. Wang *et al.* in [19] established an online algorithm to estimate social-aware reward functions, by inferring three kind of characteristics, including egoism, courtesy and confidence. Schwarting *et al.* in [20] formulated interactions between AVs as a best-response (BR) game, and then they proposed a metric called Social Value Orientation (SVO) which quantifies the continuous degree of an agent's selfishness or altruism in reward definition. This approach allows AVs to observe other traffic participants, estimate their SVOs, and generate an autonomous control policy in real time. Furthermore, estimating SVOs at consecutive timesteps can better predict how AVs will interact with others, which also inspires several solutions for several sequential social dilemma (SSD) games [21, 22]. However, existing methods focus on all of traffic participants simultaneously, which may bring increasing challenges as deploying more vehicles. Thus, in this paper, we consider how to filter redundant information by selecting the most interactive traffic participant as the interacting object for each AV.

# 3 Preliminaries

## 3.1 Dec-POMDP

We consider an MASD system where $M$ independent AVs aim to reach their given destinations quickly and safely. To apply MARL, we formulate this system as a decentralized partially observable Markov decision process (Dec-POMDP [23]), represented by a tuple $(\mathcal{M}, \mathcal{S}, \mathcal{O}, \mathcal{A}, \mathcal{R}, \Pi, \Pr, \gamma)$, where $\mathcal{M}, \mathcal{S}, \mathcal{O}, \mathcal{A}$ and $\Pi$ are the set of AVs, states, local observations, actions and policies, and $\gamma \in [0, 1]$ is the discount factor. The system works as follows, episodically. At the beginning of an episode, the global state is initialized as $\boldsymbol{s}_0$. At each timestep $t$, each AV $m \in \mathcal{M}$ has its own observation $\boldsymbol{o}_t^m$ of state $\boldsymbol{s}_t$, including kinematic information of itself and other observable AVs. Then, it decides an action $\boldsymbol{a}_t^m$ sampled from its policy $\boldsymbol{a}_t^m \sim \pi^m(\cdot|\boldsymbol{o}_t^m)$. After the environment receives all AVs' joint action (i.e., $\mathbf{a}_t = \{\boldsymbol{a}_t^m\}_{m=1}^M$), each AV $m$ receives the individual extrinsic reward $r_t^m = \mathcal{R}^m(\boldsymbol{o}_t^m, \boldsymbol{a}_t^m)$, which evaluates its performance at timestep $t$; followed by transiting to the next state $\boldsymbol{s}_{t+1}$, based on the state transition distribution $\Pr(\boldsymbol{s}_{t+1}|\boldsymbol{s}_t, \boldsymbol{a}_t)$. The episode will be terminated if: (a) all of the AVs have arrived at their destinations, (b) some of them collide with each others. For each AV $m$, the optimization problem is to find an optimal policy $\pi^m \in \Pi$ to maximize the accumulated discounted reward, as:

$$\begin{aligned} \max_{\pi^m} \quad & \mathbb{E}\big[ \sum_{t=1}^{\infty} \gamma^t r_t^m(\boldsymbol{o}_t^m, \boldsymbol{a}_t^m) \big] \\ \text{s.t.} \quad & \boldsymbol{o}_t^m \in \mathcal{O}, \; \boldsymbol{a}_t^m \in \mathcal{A}, \; \pi^m \in \Pi. \end{aligned} \tag{1}$$

Note that each AV $m \in \mathcal{M}$ optimizes its policy $\pi^m$ independently in our formulated Dec-POMDP.

## 3.2 Independent Control by Proximal Policy Optimization

To solve this Dec-POMDP, a simple approach is to allow each AV to maximize its individual objective independently, as in the single-agent problems. The difference is that each AV only utilizes its observation $\boldsymbol{o}_t^m$ instead of the full state $\boldsymbol{s}_t$. Extended by the policy-based method PPO [24], the value function and advantage function of AV $m$ at timestep $t$ can be computed as:

$$V_t^m = V^m(\boldsymbol{o}_t^m) = \mathbb{E}\Big[ \sum_{t=1}^{\infty} \gamma^t r_t^m \Big], \tag{2}$$

$$A^m(\boldsymbol{o}_t^m, \boldsymbol{a}_t^m) := r_t^m + \gamma V_{t+1}^m - V_t^m, \tag{3}$$

Then, by adopting a truncated importance sampling factor $\rho = \frac{\pi_{\text{new}}^m(\boldsymbol{a}^m|\boldsymbol{o}^m)}{\pi_{\text{old}}^m(\boldsymbol{a}^m|\boldsymbol{o}^m)}$, the surrogate objective of PPO is written as:

$$J_{\text{PPO}}(\pi^m) = \mathbb{E}[\min(\rho A^m, \text{clip}(\varrho, 1 - \epsilon, 1 + \epsilon)A^m)], \tag{4}$$

where clip function helps restrict the incentive of optimizing $\pi^m$ towards $r_t^m$ within the interval $[1 - \epsilon, 1 + \epsilon]$. Note that $\epsilon$ is a real number and specify the interval. The value function $V^m$ is updated by mean squared error loss function:

$$\mathcal{L}^m = \mathbb{E}[r_t^m + \gamma V^m(\boldsymbol{o}_{t+1}^m) - V^m(\boldsymbol{o}_t^m)]^2. \tag{5}$$

# 4 Methodology

We propose a novel MARL algorithm called Socially-Attentive Policy Optimization (SAPO) to achieve better cooperative patterns in MASD systems. As shown in Fig. 2, SAPO uses PPO as the start point of the design, and consists of two improvements: a self-attention module called "Interactive Attention" to select the most interactive participant for each AV from complex traffic flows (see Section 4.1), and a Social-aware Integration mechanism to model the impacts of SPs in vehicle interaction (see Section 4.2). For simplicity, we omit the timestep index $t$ in this section. The whole structure is depicted by the following equations (given $\forall m \in \mathcal{M}$):

$$(p, \boldsymbol{h}^m) = \text{InteractiveAttention}(\boldsymbol{o}^m), \quad p \in \mathcal{M} \backslash m, \tag{6a}$$

$$r_{\text{SP}}^m = \text{Integration}(r^m, r^p|\boldsymbol{o}^m, \boldsymbol{h}^m) \tag{6b}$$

$$\pi^m(\boldsymbol{a}^m|\boldsymbol{o}^m, \boldsymbol{h}^m) \equiv \text{f}^\pi(\boldsymbol{o}^m, \boldsymbol{h}^m), \tag{6c}$$

$$V^m(\boldsymbol{o}^m, \boldsymbol{h}^m) \equiv \text{f}^V(\boldsymbol{o}^m, \boldsymbol{h}^m), \tag{6d}$$

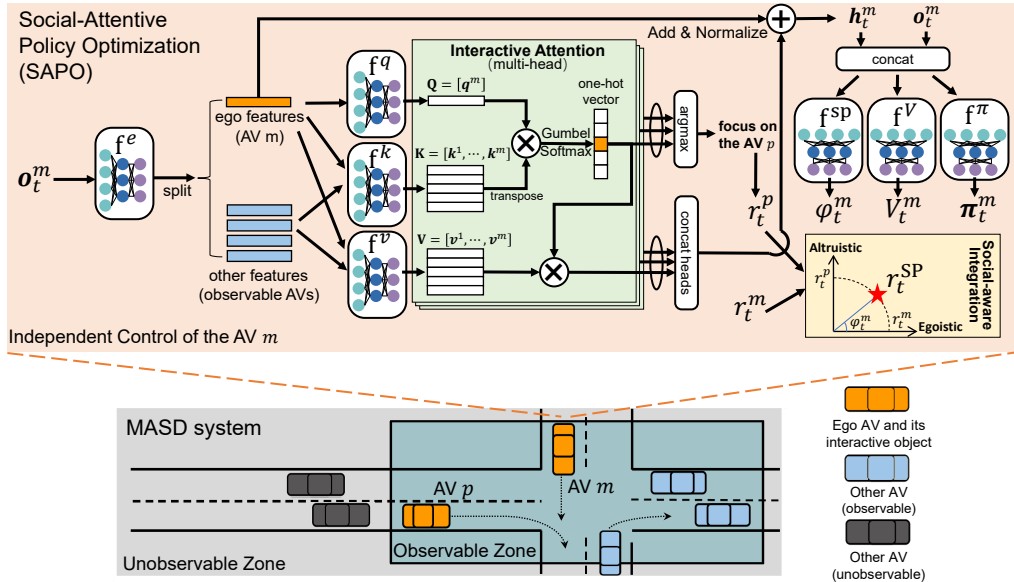

Figure 2: Architecture of SAPO.

where $p$ is the index of the traffic participant which is most likely to interact with AV $m$; $r_{SP}^m$ represents the SP-aware reward integrated from interacting AVs; $f^\pi$, $f^V$ are multi-layer perceptrons (MLPs). For each AV $m$, SAPO is running as follows. First, SAPO utilizes Interactive Attention to select the most interactive participant $p$ and generate a compact feature $h^m \in \mathbb{R}^n$ (see Eqn. (6a)). Then, the individual reward $r^m$ is replaced by an SP-aware reward $r_{SP}^m$, through Social-aware Integration mechanism (see Eqn. (6b)). Next, through $f^\pi(\cdot)$ and $f^V(\cdot)$, the compact feature is mapped into an action distribution $\pi^m$ and a value function $V^m$, respectively (see Eqn. (6c) and Eqn. (6d)). Finally, similar to PPO, the advantage function of AV $m$ is computed by Eqn. (3), for PPO updates by Eqn. (4) and Eqn. (5). Other Details of running SAPO are deferred in Appendix A.

## 4.1 Interactive Attention for Selecting Interacting Objects in Traffic Flows

Since there are usually numerous AVs in MASD system, it is necessary for each AV to filter the observed information and consider what is relevant for decision. We focus on the fact that each AV is mostly affected by its nearby agents rather than focus on all of the others simultaneously [25]. That is, each AV should pay attention to the most interactive vehicle that is close or conflict with the planned route. Inspired by [26], we propose a self-attention architecture called "interaction attention". It is used to both select the most interactive participant and generate a compact feature.

As shown in Fig. 2, for each AV $m$, SAPO splits $o^m$ into the observation of the ego vehicle and the others at first, which are embedded into $M$ vectors $x$ of size $d_x$, by a linear encoder $f^e$. Note that each AV has its own observable zone and the feature vectors of AVs outside this zone will be set to zeros. Through MLP $f^q$, $f^k$ and $f^v$, all of the $M$ vectors are embedded into the queries ($q$), keys ($k$) and values ($v$) of size $d_k$. They are then fed to the multi-head attention layer [27], composed of several heads stacked together. The difference is that we only use a single query $\mathbf{Q} = [q^m] \in \mathbb{R}^{1 \times d_k}$ by features of the ego AV, which is compared to all of keys $\mathbf{K} = [k^1, \cdots, k^M] \in \mathbb{R}^{M \times d_k}$, containing observed features of each AV. The similarity between the query $q^m$ and any AV's key $k$ is accessed by their dot product $q^m k^\top$ and then scaled by $\frac{1}{\sqrt{d_k}}$. Next, in order to get the most interactive participant as the interacting object, we transform the attention weights into the one-hot vector by utilizing a Gumbel Softmax [28] function $\sigma(\cdot)$. With this hard attention scheme, the index of the only non-zero element corresponds to the index of an interactive AV. We utilize the one-hot vector to gather all of the values $\mathbf{V} = [v^1, \cdots, v^M] \in \mathbb{R}^{M \times d_k}$ and get the output. Overall, the attention computation for each head can be written as:

$$\text{output} = \underbrace{\sigma\left(\frac{\mathbf{Q}\mathbf{K}^\top}{\sqrt{d_k}}\right)}_{\text{attention matrix}} \mathbf{V}. \tag{7}$$

Finally, the outputs from all heads are combined, generating compact features. Without loss of generalization, we get the final social-aware feature $\boldsymbol{h}^m \in \mathbb{R}^n$, by adding resulting tensor to the ego encoded observation of AV $m$ as in residual networks. Similarly, we also combine the attention matrix of all heads to get the most interactive participant $p$ which the AV $m$ should focus on temporally. The advantages of the Interaction Attention are: (a) inputs can have a variable size; and (b) outputs are permutation invariant which do not depend on the ordering of surrounding AVs.

### 4.2 Social-Aware Integration of Individual Objectives with Social Preferences

In an MASD system, each AV has its individual objective. However, if we simply maximize each individual reward by PPO, the system will have sub-optimal solutions where, for example, AVs will become aggressive and egocentric, jeopardizing the performance. On the contrary, if we apply fully cooperative learning schemes [11, 29] which consider the summation of individual reward as the joint objective, the trained agents will exhibit unreasonable behaviors such as sacrificing oneself to improve group reward, which is not expected in MASD systems. To find a balance, we consider utilizing the impacts of SP on learning reasonable behaviours. Thus, inspired by social psychology, we propose a social-aware integration mechanism to integrate objectives of interacting AVs by their SPs. Different from the previous *Local Coordination Factor* [15] which is fixed at each timestep and adjusted by the total reward of each active agent episodically, SAPO only utilizes the AV $m$'s observation $\boldsymbol{o}^m$ to update SP $\varphi^m$ at each timestep $t$, without any centralized training schemes.

For each AV $m$, the mechanism works as follows, as shown in the right part of Fig. 2. Since social-aware features $\boldsymbol{h}^m$ are extracted by Interactive Attention, an MLP $\mathrm{f}^{\mathrm{SP}}$ is then utilized to generate the ring measure of SVO [20] to represent the AV $m$'s SP $\varphi^m$, as:

$$\varphi^m(\boldsymbol{o}^m, \boldsymbol{h}^m) \equiv \mathrm{f}^{\mathrm{SP}}(\boldsymbol{o}^m, \boldsymbol{h}^m); \quad \varphi^m : \mathcal{O} \times \mathbb{R}^n \mapsto [0, \pi/2]. \tag{8}$$

Next, we utilize $\varphi^m$ to control the appropriate level of coordination between the AV $m$ and its interactive participant $p$, by calculating a social-aware reward $r_{\mathrm{SP}}^m$ as:

$$r_{\mathrm{SP}}^m = r^m \cos \varphi^m(\boldsymbol{o}^m, \boldsymbol{h}^m) + r^p \sin \varphi^m(\boldsymbol{o}^m, \boldsymbol{h}^m), \tag{9}$$

which replaces the AV $m$'s individual reward $r^m$ as a new objective. Different from automatic search of the optimal SP fixed in an episode [15], SAPO can estimate $\varphi^m$ at each timestep $t$ and make sure that $r_{\mathrm{SP}}^m$ is corresponding to the current traffic conditions, which is important in real scenarios.

## 5 Experiments

In this section, we test the ability of SAPO to learn reasonable behaviors and explore better policy. We first study didactic examples inspired by SSD games [30] to investigate the effects of Interactive Attention and Social-aware Integration on the learning process (see Section 5.1). To demonstrate scalability on complex MASD systems, we also evaluate the performance of SAPO on a range of SMARTS benchmark tasks [4] (see Section 5.2). Furthermore, we also discuss the applicability of SAPO in real-world autonomous driving system based on open datasets like Waymo [31] and Argoverse [32] (see Appendix D). Through the distributed training architecture supported by RLlib [33], we set different random seed in each actor to adapt a diverse set of environmental settings. Then, we evaluate the trained policies in 10 random seeds and get the maximum, mean and minimum results to visualize the results in all experiments. The detailed settings are introduced in Appendix E.

### 5.1 Toy Example: Bottleneck

Inspired by [22, 30], we design a toy example of MASD systems, called Bottleneck. Consider a grid world with 4 AVs (see Fig. 3(a)), 5 available actions (i.e., stop or move left, right, up and down) and observation space containing 2-D position of ego AV and the others. Each AV's reward is determined by the distance to its own destination and the time costs. Given the initial positions and destinations, AVs should learn to make ways for others and reach the goals in orders spontaneously. Since the solution space is limited, we can easily explore the overall optimal policy and examine the exploitation of MARL. From Fig 3(b), we observe that SAPO trains much faster than PPO and achieves the optimality. This is because AVs only see others as parts of the environment. They are likely to learn selfish behaviours (e.g., keeping closer to the goal without yielding any space) and resulting in congestion, which need to consume more time on exploring the optimal policy. Besides,

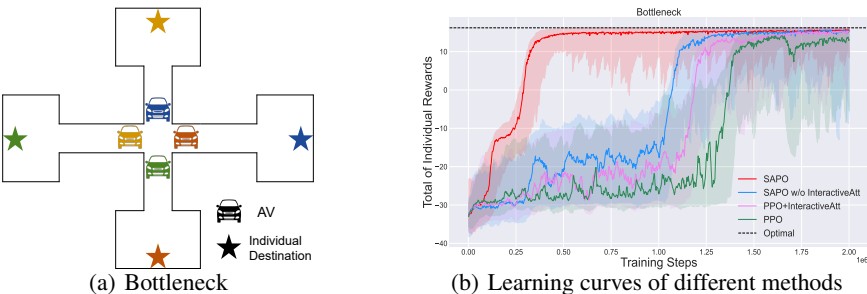

(a) Bottleneck         (b) Learning curves of different methods

Figure 3: (a) A special toy example called Bottleneck, and each AV aims to reach the destination point with the same color as soon as possible. (b) The learning curves of SAPO and other baselines.



(a) Cross-road    (b) T-junction    (c) Lane Merging    (d) Roundabout    (e) On-ramp

Figure 4: Examples of MASD scenarios in SMARTS [4].

the learning process will be much slower without Interactive Attention. This is because it is hard to maintain the efficiency and accuracy of estimating SPs when considering all of AVs together.

## 5.2 SMARTS

To tackle settings that more closely represent real world settings, we focus on MASD scenarios based on the Scalable Multi-Agent RL Training School (SMARTS) benchmark. Fig. 4 describes the examples of tasks we use in our experiments. The experimental settings are deferred to Appendix B. To demonstrate the overall performance of each algorithm, we utilize the *average success rate* as the evaluation metric, calculated by the average ratio of the number of vehicles that reach their goals over the total number of vehicles in one episode.

### 5.2.1 Ablation Study

To show the importance of utilizing Interactive Attention and modeling SPs on optimizing MASD systems, we perform the ablation study by: (a) removing Interactive Attention (i.e, calculate $r^p$ by the average reward of every surrounding vehicle) or using other mechanisms to select $p$ (i.e., replace Interactive Attention by: random attention or a rule of selecting the nearest vehicle), and (b) using given SPs instead of dynamic estimation for training SAPO in a cross-road scenario. Results are shown in Fig. 5. We see that the complete version SAPO has higher success rate than SAPO w/o InteractiveAtt. This is because Interactive Attention can filter information from a number of AVs in the intersection, which accelerates the training process. Besides, there is an intuitive finding that the rule-based attention is a powerful baseline in MASD systems, which is corresponding to some conventions of human behaviors On the other hand, we see that SAPO ($\varphi = 30°$) performs worse than SAPO ($\varphi = 45°$) and SAPO ($\varphi = 60°$). This is because SAPO ($\varphi = 30°$) encourages more self-interested AVs (according to Eqn. (9)), which usually get blocked as they refuse to make way for others. However, even if we set $\varphi = 45°$ or $\varphi = 60°$ to keep the AVs prosocial or altruistic, SAPO is still trapped into another local optimal. We observe that only the dynamic modeling of SPs can navigate AVs to learn reasonable social-aware behaviours, which improves the overall performance of the MASD system.

### 5.2.2 Illustrative Visualization of the Trained Behaviors

In Fig. 6, we show the visualization results of AVs' positions, attention weights and SPs (from the bottom to the top) at each timestep in a cross-road scenario, where each AV follows its own waypoints, navigated by SAPO independently. To predict SP smoothly, we discretize $\varphi_t$ at each timestep, as: $\varphi_t \in \{0°, 15°, 30°, 45°, 60°, 75°, 90°\}$ To better explain the learned social-aware

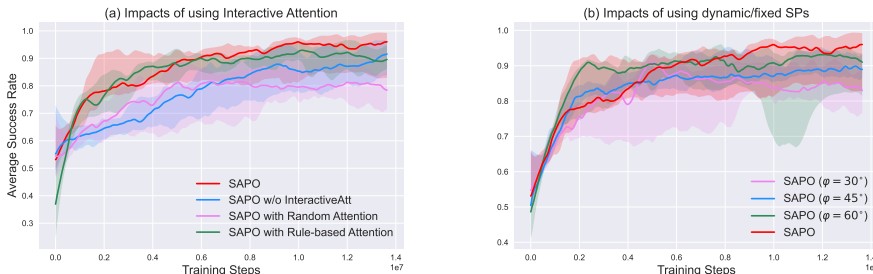

Figure 5: Ablation study.

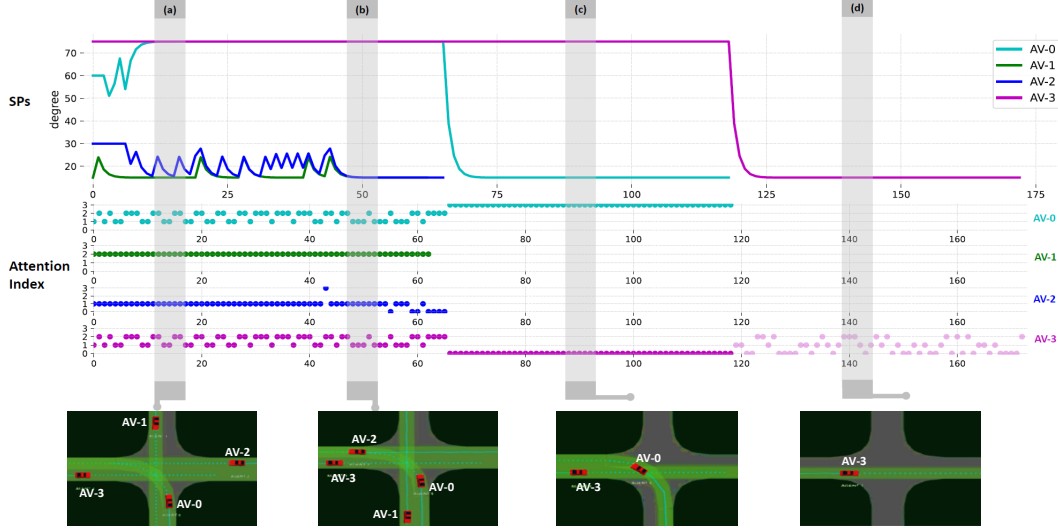

Figure 6: Illustrative visualization of AV's behaviours trained by SAPO.

behaviors, we focus on three stages in an episode. In stage $a \rightarrow b$, AV-0 and AV-3 slow down and make way for AV-1 and AV-2 to cross the intersection. Then, in stage $b \rightarrow c$, AV-0 speeds up while AV-3 stops and waits. Finally, AV-3 speeds up and crosses the road in stage $c \rightarrow d$. From the learned SPs of each AV, we observe that AVs with higher SPs can learn to make way for AVs with lower SPs. Furthermore, we observe that Interaction Attention is helpful by selecting the most relevant traffic participant. For example, when AV-0 and AV-3 both slow down, AV-1 and AV-2 only pay attention to each other when crossing the intersection. These results indicate that AVs can choose suitable vehicles as interacting objects and then learn reasonable behaviors.

### 5.2.3 Comparison Results of Training in MASD Scenarios

We evaluate SAPO in 6 popular MASD tasks. Other than PPO, we further compare SAPO against three popular baselines in self-driving scenarios as follows:

- *Iterative Best Response with SVO* [20]. As a popular game-theoretic approach for self-driving, it models the interactions between AVs as a multi-player BR game wherein each AV negotiates to maximize its own reward and predict other AVs' behaviors by estimating their SVOs in real time. The details of BR and SVO estimation are deferred to Appendix C.

- *Two-player Best Response* [34]. It trains the Interactive Attention to select the interacting object for each AV, and then finds the Nash equilibrium as in two-player BR games.

- *Social Attention* [26]. Based on the centralized control of multiple AVs, it directly utilizes a soft-attention-based architecture for decision-making involving social interactions. That is, it focus on all of surrounding vehicles with different attention weights. To remove the impacts of the control method, we also consider the variant of decentralized control, where each AV maintains its own attention and PPO framework independently.

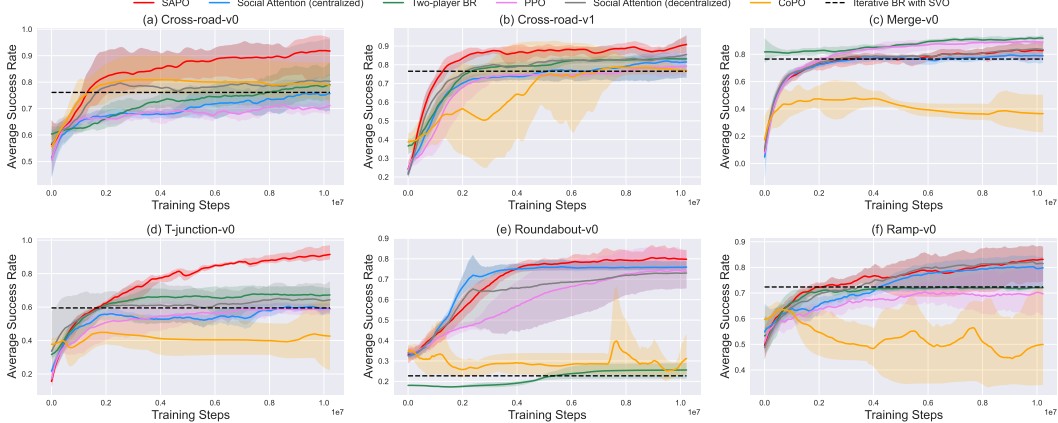

Figure 7: Training curves of SAPO and baselines in SMARTS.

- *CoPO* [15]. It proposes a state-of-the-art MARL method to navigate traffic of AVs in MASD systems, by updating PPO policies to maximize a social-aware reward like Eqn. (9). The difference is that it allows each AV to focus on all of its surrounding vehicles and utilize the average reward of them as $r^p$. Besides, it fixes $\varphi$ at each timestep and adjusts it episodically by optimizing the total reward of each active agent in the simulation. That is, CoPO is mainly used for centralized training and decentralized execution of multiple AVs.

The learning curves are shown in Fig. 7. Note that we use the testing performance of Iterative BR with SVO [20] as a reference line, since it is an online optimization-based method. We observe that, SAPO significantly outperforms baselines in most of the cases and PPO performs the worst. With attention-based feature extraction, Two-player BR and Social Attention both achieve reasonable performance, especially in lane merging scenarios where two AVs can choose when to merge so their conflicts are less than other scenarios. However, in two cross-road scenarios where AVs need to deal with more conflicts of individual goals, the performance gap between SAPO and other baselines becomes larger. This indicates that Social-aware Integration is important in learning reasonable behaviors and improving the efficiency of MASD systems. CoPO cannot perform very well because the coordination or conflicts of the AVs' goals seldom keep the same in a whole episode. Besides, it utilize the global reward and focus on all of agents together, which will bring redundant information.

# 6 Conclusion

Inspired by human cooperation in societies, this paper proposes a novel reinforcement learning method called SAPO to learn social-aware controller of MASD systems, which incorporates a self-attention module and social-aware integration mechanism for selecting interacting objects and integrating individual objectives of AVs, respectively. Experiments on basic traffic simulation show that SAPO can successfully estimate the SPs from AVs' observations and learns social-aware behaviours, which achieves better performance than baselines in several popular MASD systems.

# 7 Limitations and Future Research Directions

This paper has several limitations. First, we focus on decision-making problems, so we simplify the perception of AVs and assume the observation is based on accurate sensory data, which is not easy to achieve in reality. Second, we only utilize the current observations to predict SPs and make decisions, without referring to some trajectory prediction works [16, 17, 18] which make efficient use of historical trajectory data. Third, we only maintain the trial-and-error process in diverse environmental settings, by training within multiple random seeds together. There should be some better ways which achieve zero-shot transfer to unseen partners without using too much experience data. Finally, we adopt $\varphi \in [0, \pi/2]$ in this paper and there should be more diverse scenarios (e.g., car racing) where SP can be defined beyond $[0, \pi/2]$. How to tackle with these limitations is a promising research direction.

**Acknowledgments**

The authors would like to thank the anonymous reviewers for their helpful comments and thank the many Huawei colleagues in China and UK.

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
