# OpenReview forum: "Socially-Attentive Policy Optimization in Multi-Agent Self-Driving System"
_robot-learning.org/CoRL/2022/Conference — CoRL 2022 Poster_

### Official Review · Reviewer_6Syb · 2022-07-12

**Originality:** Good
**Technical Quality:** Good
**Clarity Of Presentation:** Good
**Impact:** 3

**Recommendation:**

Weak Accept: I recommend accepting the paper, but will not argue for my recommendation if the majority of other reviewers have a different opinion.

**Summary:**

The article presents SAPO: a modified version of the IPPO algorithm to solve AV interaction stated as a Multi-Agent Reinforcement Learning problem. The first modification is using social attention (Leurent and Mercat, 2019) to select *one* "most interactive" agent. The second modification is estimating a dynamic "social preference" (aka Social Value Orientation, Schwarting, et al., 2019) which combines rewards from ego and interactive agents to form a new reward for ego.
The article shows a favorable comparison of SAPO performance against a set of baselines, using the total of individual rewards as the metric.

**Issues:**

1. Addressing the Weakness 1. Perhaps the motivation can be changed: e.g. simulating self-driven particles for improving traffic simulation as in Peng et al., 2021; this would however require changing of metric as well. Alternatively demonstrating that the SAPO agents will be able to deal with a diverse set of other agents. There might be other possibilities as well.
    * It would be very useful to underline relevance to robotics, e.g. if you could show the perspective of integrating this into a real-world autonomous driving system.

2. Explain in more detail, how the method SAPO without Interactive Attention works. Without the Interactive Attention, there is no AV p, so whose reward is used in computing social-aware reward (Eq. 9)?

3. Provide more details on the social attention baseline. If possible, create another baseline, where social attention is used in a decentralized setup.

**Quality Of The Limitations Section:**

Additional details required

**Reviewer Expertise:**

4: The reviewer is confident but not absolutely certain that the evaluation is correct

**Robotics Focus:**

Highly relevant to robotics but no hardware experiments

**Strengths And Weaknesses:**

Strengths:
1. The presented way of combining recently developed techniques (social attention, SVO, IPPO) seems to be novel, and potentially promising.
1. The set of baselines is large and diverse (perhaps not comprehensive though, see weakness 2).
1. The article presents in a clear way the methods it uses (IPPO, social attention).
1. Fig. 6. is a very nice qualitative presentation of a particular result.

Weaknesses:
1. In my opinion, the main problem of this work is that the motivation does not match with the proposed method and evaluation. The motivation is that AVs are becoming more popular and are sharing the space with other traffic participants (lines 23-24), so safe and smooth interactions with other agents are required (line 34). The solution is a Multi-Agent Self-Driving system, where all agents are trained in the same training loop. Even if such a multi-agent system achieves a high total reward (the metric used by authors), it does not guarantee that the agents would be compatible with human drivers or with different AVs.
    * The work uses simulator SMARTS, which supports "diverse behavior models of road users". Proving a good performance of agents trained with SAPO in interaction with SMARTS provided agents could address the issue to some extent.
1. The method makes a surprising assumption, that the "AV should pay attention to *the most interactive* vehicle", and the way it uses interaction attention is by selecting only one agent for the downstream processing. I think it is easy to imagine a situation, where the ego should take into account interaction with more than one agent.
     * Perhaps the point above is mitigated by including the baseline with "social attention". The method is however described with very few details. It does mention that it is based on "centralized control of multiple AVs". Maybe the lower performance is caused not by using full social attention (rather than just the most relevant), but rather due to centralized control (which is not always better).

**Summary Of Recommendation:**

In its current form, the paper does not have a good match between the motivation and the method, as well as its evaluation. Moreover, I don't see how this method could be used in a practical setup of an actual autonomous vehicle (as it assumes that other traffic participants are trained together with ego). This limits the applicability to robotics, which is the scope of the conference. While the work presents a new way of combining recently developed techniques (social attention, SVO, IPPO), the overall novelty is limited.
Unfortunately, in its current form, I recommend rejecting the article.

----

The additional experiments added during the discussion period (in particular on Argoverse and Waymo Dataset) have convinced me to increase my score to weak accept.

---

> ### Author Response · Authors · 2022-08-22
> **Response to the reviewer 6Syb**
>
> We thank the reviewer for the insightful suggestions. We address the reviewer's comments and questions individually as follows, and we have also uploaded an revised version of our paper which now includes the supplementary material temporally. For the ease of reading, we have elaborated the details of the updates in our note to all reviewers. We hope that our response will enable the reviewer to appreciate the clarity of the paper and motivation as the other reviewers have.

---

> ### Author Response · Authors · 2022-08-22
> **Response to the reviewer**
>
> #### Comment 1:
> > "Perhaps the motivation can be changed: e.g. simulating self-driven particles for improving traffic simulation as in Peng et al., 2021; this would however require changing of metric as well. Alternatively demonstrating that the SAPO agents will be able to deal with a diverse set of other agents. There might be other possibilities as well.".
> #### Answer:
> We greatly appreciate the reviewer’s effort in providing suggestions to improve the paper. We use a more common metric called “average success rate” to evaluate performance of the trained models in all of our self-driving experiments, instead of the abstract reward given by SMARTS. We calculate the metric by using the average ratio of the number of agents that reach their goals over the total number of agents in one testing episode. We also add the comparison with Peng's CoPO in SMARTS (see Figure 7 in our revised paper). We charify the main differences between CoPO and SAPO in Section 5.2.3, and here we will explain them in details:
> 1. In terms of choosing the interacting partners, CoPO focus on all of the surrounding vehicles within "a fixed range" (need to be tuned carefully as an extra hyperparameter), while SAPO focus on one of the surrounding vehicles by using Interactive Attention. We have discussed the advantages of Interactive Attention in our ablation studies and comparison with Social Attention (decentralized).
> 2. CoPO fixes the SP of each AV at each timestep and adjusts it episodically by using the total reward of all active agents in Metadrive. Differently, SAPO updates the SP step by step, by only utilizing the current observation of the ego AV. However, considering multiple training cases together, we think the coordination or conflicts of the AVs' goals will not keep the same in a whole episode. This is the main reason why CoPO cannot perform very well in some scenarios (see Figure 7).
> 3. From the previous points of differences, it can be concluded that CoPO is mainly designed for centrailized training and decentralized execution (CTDE) of multiple AVs. For MASD systems, it is an interesting and powerful framework in "multi-agent control", where AVs should be trained together. However, without the dependence of the global information, SAPO can not only work in "multi-agent control" (by training multiple AVs independently), but also in "single-agent control" (by training a single AV to interact with social vehicles). We have discussed how to apply SAPO in a single-agent self-driving environment (see Section D of the Appendix).
>
> Besides, we give more implementation details about the randomness of the training cases in SMARTS, and how to deal with it by using the distributed RL architecture like RLlib (see Section A and B in the Appendix). We address that the randomness of the simulation environment is an important issue for RL training. If the initial state keeps the same during the training and testing process, it is easy to search how to achieve the optimal policy with 100% success rate, which is far from real-world application. To this end, **we utilize a random seed to represent a unique permutation of several important settings**, including: a) the entrance from which the vehicle enter; b) the initial speed of the vehicle; and c) the given route which the vehicle is following, such as moving straight, changing lanes and turning left/right. We choose several valid settings where vehicles obey the road rules and save the relevant seeds as the training and testing cases. For implementation that matters, we utilize RLlib, a powerful distributed training architecture for RL training. To make sure our trained policy can adapt a diverse set of environmental settings, we set different random seed in each actor.
>
> Actually, using random seed as different training/testing cases is common in many simulation environments (e.g., Multi-agent Particle Environment, Box2D, Metadrive). However, it is still not enough to demonstrate SAPO can deal with any kind of unseen agents. Thus, we enrich the limitation section of this paper, by considering this will be a promising start point of apply RL in real-world applications.

---

> > ### Author Response · Authors · 2022-08-22
> > **Response to reviewer 6Syb continued...**
> >
> > #### Comment 2:
> > > "It would be very useful to underline relevance to robotics, e.g. if you could show the perspective of integrating this into a real-world autonomous driving system."
> > #### Answer:
> > We greatly appreciate the reviewer’s effort in providing suggestions to improve the paper. As shown in Figure 8, We evaluate SAPO in a more realistic self-driving system, where SAPO controls a single AV to interact with the real traffic based on Waymo Motion Dataset and Argoverse Dataset. Considering SAPO works as an independent controller of each AV and does not require any global information for training, we can directly test SAPO in this single-agent control scenario, without major modifications. In Appendix, we have shown some potential applications. However, we also find some open questions of replaying the RL-based vehicles and data-driven vehicles together in an environment. More details are given in Section D of the Appendix.
> >
> >
> > #### Comment 3:
> > > "Explain in more detail, how the method SAPO without Interactive Attention works. Without the Interactive Attention, there is no AV p, so whose reward is used in computing social-aware reward (Eq. 9)?"
> > #### Answer:
> > We thank the reviewer for this comment. We have explained more details in Section 5.2.1 of our revised papers. Without Interactive Attention, an AV should focus on all of its surrounding vehicles and utilize the average reward of them as $r^p$ to compute social-aware reward (like Peng et al. in NeurIPS'2021 and Schwarting et al. in PNAS'2019)
> >
> >
> > #### Comment 4:
> > > "Provide more details on the social attention baseline. If possible, create another baseline, where social attention is used in a decentralized setup."
> > #### Answer:
> > We thank the reviewer for this comment. We have provided more details on the social attention baseline, and add another baseline of social attention used in decentralized control for fair comparison (see Section 5.2.3 of our revised paper.)

---

> > > ### Comment · Reviewer_6Syb · 2022-08-28
> > > **Response**
> > >
> > > Thank you very much for the detailed answer to my review. With the additional experiments and clarifications, I'm inclined to raise the score to weak accept.
> > >
> > > The experiment with real-world data (presented in Appendix D) were particularly important to me. For the future, I'd suggest exposing them more also in the main body of the article.

---

### Official Review · Reviewer_JLzw · 2022-07-25

**Originality:** Very Good
**Technical Quality:** Very Good
**Clarity Of Presentation:** Very Good
**Impact:** 3

**Recommendation:**

Weak Accept: I recommend accepting the paper, but will not argue for my recommendation if the majority of other reviewers have a different opinion.

**Summary:**

This paper presents SAPO, a multi-agent RL framework for autonomous driving. SAPO includes an attention module that tries to identify the most important "other" car in an interactive driving scenario and then compactly represent the features of this interaction. It then attempts to integrate these features in such a way to dynamically update the reward structure for the agent, notionally capturing social preferences and adaptation. The paper presents an ablative study, attempting to isolate the effects of several design choices. It then presents an evaluation across several scenarios, compared to 3 popular baseline models. The paper is largely well-written, and the idea is conceptually clever and very important. I have described several concerns below and hope they are instructive and helpful.

**Issues:**

Line 9, abstract: "or" instead of "of" in cooperative or competitive fashion...

Line 27: "self-driven particles"? What is a particle here?

Line 39-40: "most of the interaction events occur in the red zone when AVs are crossing the intersection". While I agree that intersections include a lot of interactions, it is unclear that "most" of the interaction events occur here.

The authors do well to review the literature on social preferences in section 2.2. However, there is another, broader literature on social preferences and context in navigation that could (should) be explored here, for example how human pedestrians navigate in crowded settings. For example, Sekhon and Fleming. (2021, May). SCAN: A Spatial Context Attentive Network for Joint Multi-Agent Intent Prediction. In Proceedings of the AAAI Conference on Artificial Intelligence. and many others from Alexandre Alahi et al.  The dynamics of pedestrians (or ships, etc) are obviously quite different, but the attention mechanisms are likely quite similar and this body of literature could be helpful.

Line 92 we see "particles" again. Is this a common term? Does it imply that these AVs do not have kinematics or something?

In lines 126 onward the authors intentionally omit the timestep in the notation for simplicity. In my opinion, this makes the text and equations more difficult to interpret.

What is the justification for equation (9)? Why sine and cosine of the social preference? Why addition? Is this tailored to intersections? What about other types of interactions (e.g. merging or passing in multi-lane traffic)? Would these types of interactions require different integrative rewards?

In general, I like the ideas and notions presented in section 4. The ability to focus one's attention and adapt rewards accordingly is both clever and important. However, I am a bit concerned about the lack of justification for the particular choices, and then a lack of alternatives and/or comparisons to state of the art in the ensuing sections. These design choices either need a theoretical justification or empirical evaluation, or better yet both. There are ablative studies, which is at least a start in the right direction.

The ablation studies (and other evaluations) are presented in terms of total rewards. This is a fine place to start, but it assumes that the individual rewards are themselves representative of the desired task. What about (just making some metrics up here) collision rates? Average time to nagivate across an intersection? Or a normalized score comparing this travel time to perfect performance when no other vehicles are present? Etc etc etc. The problem with using rewards only is that we do not really know whether the RL agents have learned anything useful without any other context. Again, this is ok as part of a larger set of evaluations, but I do not see this.

I appreciate the comparison with several popular baselines, across a variety of scenarios. SAPO does well in these comparisons. In particular, it does well with the cross-road scenario, but is this due to the overall concept? Or due to particular design choices (I am thinking of equation (9) for example)? It seems like a general notion of attention and integration should be better across all interaction scenarios, and merging and ramps are perfectly valid instances of interactive scenarios but may be not-quite represented by the authors' design choices.

Line 253: "This proves...". This does not really prove anything. It demonstrates that SAPO did better than a few baselines, across a few scenarios, in terms of overall reward. It is an *indication* of *something*, which is fine and quite enough to just say this.

There are several limitations (and thus future research directions) that are not mentioned in section 7. More exploration of other attention and integration mechanisms is perhaps the most important, IMHO. Furthermore, this limitations section appears to be quite superficial.

**Quality Of The Limitations Section:**

Limitations are not well addressed

**Reviewer Expertise:**

4: The reviewer is confident but not absolutely certain that the evaluation is correct

**Robotics Focus:**

Highly relevant to robotics but no hardware experiments

**Strengths And Weaknesses:**

Strengths
- The notion that most RL agents and settings use a static reward function is important; the ability to consider dynamic, evolving rewards is a compelling one.
- The combination of social attention in a MARL setting is quite interesting, and the paper develops these notions well

Weaknesses
- Many of the evaluations are presented in terms of total accumulated reward with episodes along the x-axis. This suffers the same weakness (in my opinion) as almost all of the RL literature in that (as the very RL literature itself suggests) the reward function does not always indicate whether the task itself has been solved in a satisfactory manner.
- The limitations sections is somewhat superficial

**Summary Of Recommendation:**

The paper is conceptually very elegant, and the application area is very important (furthermore, this idea could be applied in many other settings). Although I voice my concerns below, I feel that the paper warrants publication and the community would benefit from seeing the notions of attention and MARL combined in this type of way.

---

> ### Author Response · Authors · 2022-08-22
> **Response to the reviewer JLzw**
>
> We thank the reviewer for the insightful suggestions. We address the reviewer's comments and questions individually as follows, and we have also uploaded an revised version of our paper which now includes the supplementary material temporally. For the ease of reading, we have elaborated the details of the updates in our note to all reviewers.

---

> ### Author Response · Authors · 2022-08-22
> **Response to the reviewer**
>
> #### Comment 1:
> > "Line 9, abstract: "or" instead of "of" in cooperative or competitive fashion...".
> #### Answer:
> We thank the reviewer for this comment. We have fixed this issue in our revised papers.
>
>
> #### Comment 2:
> > "Line 27: "self-driven particles"? What is a particle here?"
> #### Answer:
> A particle refers to an individual agent which pursues its own goal and interacts with each other following simple local alignment. In a multi-agent system, we use the population of particles to exhibit complex collective behaviors. To make it easier to get the key point of this paper, we delete all of the "self-driven particles" and use "AV" instead in our revised version.
>
> #### Comment 3:
> > "Line 39-40: "most of the interaction events occur in the red zone when AVs are crossing the intersection". While I agree that intersections include a lot of interactions, it is unclear that "most" of the interaction events occur here."
> #### Answer:
> We thank the reviewer for this comment. We have fixed this issue in our revised version.
>
>
>
> #### Comment 4:
> > "The authors do well to review the literature on social preferences in section 2.2. However, there is another, broader literature on social preferences and context in navigation that could (should) be explored here, for example how human pedestrians navigate in crowded settings. For example, Sekhon and Fleming. (2021, May). SCAN: A Spatial Context Attentive Network for Joint Multi-Agent Intent Prediction. In Proceedings of the AAAI Conference on Artificial Intelligence. and many others from Alexandre Alahi et al. The dynamics of pedestrians (or ships, etc) are obviously quite different, but the attention mechanisms are likely quite similar and this body of literature could be helpful."
> #### Answer:
> We greatly appreciate the reviewer’s effort in providing suggestions to improve the paper. In Section 2.2, we have added the discussion of SCAN and other similar works about trajectory or intent prediction. We think they are different from our paper because they are mainly designed for feature representation, which may be closely connected to the decision-making architectures like SAPO. We agree that reinforcement learning methods also need improvements of representation learning, like supervised learning. Thus, we also mention these works in our Section 7, as one of limitations and future research directions.
>
>
>
> #### Comment 5:
> > "Line 92 we see "particles" again. Is this a common term? Does it imply that these AVs do not have kinematics or something?"
> #### Answer:
> We thank the reviewer for this comment. To make it easier to get the key point of this paper, we delete all of the "self-driven particles" and use "AV" instead in our revised version.
>
>
> #### Comment 6:
> > "In lines 126 onward the authors intentionally omit the timestep in the notation for simplicity. In my opinion, this makes the text and equations more difficult to interpret."
> #### Answer:
> We thank the reviewer for this comment. We will think about your advice and modify all of the notations in our final version.
>
>
> #### Comment 7:
> > "What is the justification for equation (9)? Why sine and cosine of the social preference? Why addition? Is this tailored to intersections? What about other types of interactions (e.g. merging or passing in multi-lane traffic)? Would these types of interactions require different integrative rewards?"
> #### Answer:
> Overall, the representation of SP is inspired by "Social behavior for autonomous vehicles" (Schwarting et al. in PNAS'2019). It can be seen as a "Reward angle" and the ring formulation of Social Value Orientation (SVO). In physical meanings, reward angle is a scalar representation of the tradeoff between an agent’s own reward and the reward of other agents in the environment, which translates into altruistic, prosocial, egoistic, or competitive preferences. The formulation can be applied to every scenarios, but the range of SP can be different. For example, this paper mainly consider $[0,\pi/2]$ in all of MASD systems supported by SMARTS, including merging or passing in multi-lane traffic. This is because the SP in $[-\pi/2,0]$ may bring some aggressive or unsafe behaviours. However, Schwarting et al. mentioned that $[-\pi/2,0]$ may also be useful to form some competitive behaviours in car racing scenarios.

---

> > ### Author Response · Authors · 2022-08-22
> > **Response to reviewer JLzw continued...**
> >
> > #### Comment 8:
> > > "In general, I like the ideas and notions presented in section 4. The ability to focus one's attention and adapt rewards accordingly is both clever and important. However, I am a bit concerned about the lack of justification for the particular choices, and then a lack of alternatives and/or comparisons to state of the art in the ensuing sections. These design choices either need a theoretical justification or empirical evaluation, or better yet both. There are ablative studies, which is at least a start in the right direction."
> > #### Answer:
> > We thank the reviewer for this comment. We have added more ablation studies about how to choose the interacting partner in Fig. 5(a), by using other mechanisms to select $p$ (i.e., to replace Interactive Attention by random attention or a rule of selecting the nearest vehicle). From the results, we prove that using some greedy rules can improve the performance of SAPO, but still need improvements to achieve state-of-th-art performance. That is, Interactive Attention can provice an promising direction of choosing the interacting partner, before we have got a series of well-defined rules.
> >
> >
> > #### Comment 9:
> > > "The ablation studies (and other evaluations) are presented in terms of total rewards. This is a fine place to start, but it assumes that the individual rewards are themselves representative of the desired task. What about (just making some metrics up here) collision rates? Average time to nagivate across an intersection? Or a normalized score comparing this travel time to perfect performance when no other vehicles are present? Etc etc etc. The problem with using rewards only is that we do not really know whether the RL agents have learned anything useful without any other context. Again, this is ok as part of a larger set of evaluations, but I do not see this."
> > #### Answer:
> > We greatly appreciate the reviewer’s effort in providing suggestions to improve the paper. Actually, we utilze the total of individual reward because SMARTS always set great penalties to an AV if it does not reach the goal at the end of the episode or collide with other vehicles. Thus, we can see some gaps if we evaluate the total reward of effcient traffic navigation and congestions in MASD systems. However, we agree that using a abstract reward is limited to evaluate the training and test process, without enough physical meanings. To this end, we consider your comments very carefully and try to use a more reasonable metric.
> >
> > We do not consider the collision rate for two main reasons. First, in the training cases, the episode will be terminated with penalty when any AV colloides in SMARTS. This is for the purpose of safe driving, where it is difficult and meaningless to simulate other active vehicles when an accident happens. This setting is also applicable outside the scope of MASD systems. For example, in single-agent self-driving environment with real data (see Section D of the Appendix), it is also difficult to simulate the steps after the ego AV collides with several data-driven vehicles. Second, in the testing cases, we observe that AVs will usually learn to stop when they are stuck by each other, rather than collide with each other. This is because the base objective of SAPO (an RL method) is to maximize the individual reward and reduce great penalties at first. As a result, SAPO and most of its baselines can achieve nearly $0$ collision rate, without reasonable gaps.
> >
> > To this end, in the revised paper, **we decide to use a common metric called “average success rate” to evaluate the performance in our MASD systems, instead of our abstract reward**. We define the metric as the average ratio of the number of agents that reach their goals over the total number of agents in an episode. During each evalution or testing process of the policies, we simulate $20$ episodes to calculate the average success rate. Note that we have modified the y-axis of the training curves in SMARTS experiments, based on the same hyperparameters as our initial paper.

---

> > > ### Author Response · Authors · 2022-08-22
> > > **Response to reviewer JLzw continued...**
> > >
> > > #### Comment 10:
> > > > "I appreciate the comparison with several popular baselines, across a variety of scenarios. SAPO does well in these comparisons. In particular, it does well with the cross-road scenario, but is this due to the overall concept? Or due to particular design choices (I am thinking of equation (9) for example)? It seems like a general notion of attention and integration should be better across all interaction scenarios, and merging and ramps are perfectly valid instances of interactive scenarios but may be not-quite represented by the authors' design choices."
> > > #### Answer:
> > > We thank the reviewer for this comment. Actually, vehicles should learn to avoid stuck in every scenario, and we mainly highlight the performance in cross-road scenario for example. This is because the conflicts of their goals are usually more than other scenarios like Merge-v0, with less number of traffic participants and less choices of routes. In Figure 7, we can see that SAPO can work very well in most of scenarios, and the improvement compared to baselines is mainly determined by the conflicts of agents' goals.
> > >
> > >
> > > #### Comment 11:
> > > > "Line 253: "This proves...". This does not really prove anything. It demonstrates that SAPO did better than a few baselines, across a few scenarios, in terms of overall reward. It is an indication of something, which is fine and quite enough to just say this."
> > > #### Answer:
> > > We thank the reviewer for this comment. We modify the usage of "proves" in our revised paper. Besides, without loss of generality, we use average success rate instead of global reward to evaluate the performance. We also add more reasonable baselines and ablation studies.
> > >
> > >
> > > #### Comment 12:
> > > > "There are several limitations (and thus future research directions) that are not mentioned in section 7. More exploration of other attention and integration mechanisms is perhaps the most important, IMHO. Furthermore, this limitations section appears to be quite superficial."
> > > #### Answer:
> > > We thank the reviewer for this comment. To be more superfical, we consider the concerns of every reviewer to enrich the Section 7 of the revised paper, including:
> > > 1. We focus on decision-making problems, so we simplify the perception of AVs and assume the observation is based on accurate sensory data, which is not easy to achieve in reality.
> > > 2. We only utilize the current observations to predict SPs and make decisions, without referring to some trajectory prediction works which make efficient use of historical trajectory data.
> > > 3. We only maintain the trial-and-error process in diverse environmental settings, by training within multiple random seeds together. There should be some better ways which achieve zero-shot transfer to unseen partners without using too much experience data.
> > > 4. We adopt $\varphi\in[0,\pi/2]$ in this paper and there should be more diverse scenarios (e.g., car racing) where SP can be defined beyond $[0,\pi/2]$.
> > >
> > > In our opinion, how to tackle with these limitations is a promising research direction.

---

### Official Review · Reviewer_zrSF · 2022-07-31

**Originality:** Good
**Technical Quality:** Very Good
**Clarity Of Presentation:** Very Good
**Impact:** 3

**Recommendation:**

Weak Accept: I recommend accepting the paper, but will not argue for my recommendation if the majority of other reviewers have a different opinion.

**Summary:**

A method called Socially-Attentive Policy Optimization (SAPO) for individual, coordinated AV control in multi-agent driving environments, aiming to learn socially compliant behaviors between agents with different goals. Contributes two novel parts:
1. a self-attention module to select the most interactive participant (other AV) for each AV in the scene
2. a "social-aware" method to balance individual rewards (social coordination) as a balance between being purely egotistical (only ego agent reward matters) and purely altruistic (only other agent reward matters)

**Issues:**

## Larger points:
 - section 2: what about other works like Multiple Futures Prediction and Trajectron? Multi-agent autoregressive method with categorical latent variables can encode different types of behaviors or preferences too, and are (in addition) functions of the full history of past behaviors, while still reacting to latest observation (compared to this work which only looks at latest observation).
 - equation 6c, 6d: should these be function of p?
 - line 171: "unreasonable behaviors such as sacrificing oneself to improve group reward", why is this bad? Isn't this just \phi=\pi/4 in equation 9? Especially when lines 245-246 say that summed group reward is the evaluation criteria anyway.
 - section 5.1: while SAPO trains 5x faster than IPPO, how do we know this isn't just poor hyperparameter choice for IPPO? It asymtotes the same. Also, what are the uncertainty regions on Figure 3, what do they represent?
- lines 183-184: mention "corresponding to the latest traffic conditions" but this seems to be overselling: it's *only* a function of the current observations. There is no history...
 - line 211: "(a) removing Interactive Attention": a more interesting ablation here would have been *random* Interactive Attention. This would isolate-away the selection of p, and test how good the socially-aware part of the method would be. Possible an indication of how "toy" these experiments are would be: is the performance just the same? I imagine it could be, but for any sufficiently complicated setup  (perhaps what is shown in figure 6, or some more complicated traffic simulator) the ablation would show just how necessary it is to selection p intelligently.
- Why does each vehicle only select the most interactive other vehicle? there's no reason it can't focus on all vehicles, similar to [30], which inspired this work?
- Clarity: I think reading the abstract and introduction, it was difficult to understand what was actually being proposed. There were mentions of abstract benefits like "social preferences", "social-aware integration mechanism to integrate objectives of interacting AVs", and even "dynamic social preferences", and "socially-compatible behaviors", but it'd be good to get to the point faster. At line 54 it's still unclear what "socially-compatible behaviors" means, also on line 73 and 76, would help clarify to define this concretely earlier.

## Small points:
 - line 23: "inreasing" --> "increasing"
 - line 63" "MARL is widely used to navigate AVs in MASD systems" perhaps clarify this means the research litreaeture, not real AVs? Hardly any real AVs use RL, let alone MARL.
 - line 72: "response" --> "responses"
 - line 95: why is the policy space \mathcal{U} and not \Pi?
 - equation 3: doesn't the advantage function usually (1) not take a reward as input and (2) is a function of the *expected* reward as Q-V
 - equation 6a: "p \in \mathcal{M}" --> "p \in \mathcal{M} \ \{m\}"? I.e. I assume "m" cannot itself be its own closest/interactive neighbor?
 - line 159: "features" --> "generalization" (Wlog?)
 - Figure 2 (yellow box): it is shown how r^SP: goes between egotistic (\phi=0) and alteristic (\phi=\pi/2). But equation 8 mentions the range can go to -pi/2 radians too. Is this intended? I.e. ego acting "maliciously", against the other vehicles interests? If so, should it be shown in Figure 2 as well, or should the range of \phi in equation 8 be limited to [0, \pi/2]?
 - lines 174-176: different how? perhaps \emph{} which specific part following "[20]" that is different?
- lines 197: "goals in orders": which order? does one *particular* car need to go first?
- figure 6: maybe use white background, and zoom in more.
- line 231: who were vehicles AV-0 and AV-3 each selecting, was it respectively AV-2 and AV-1?

**Quality Of The Limitations Section:**

Limitations are addressed clearly

**Reviewer Expertise:**

3: The reviewer is fairly confident that the evaluation is correct

**Robotics Focus:**

Highly relevant to robotics but no hardware experiments

**Strengths And Weaknesses:**

Strength:
 - Learning 5x faster in some cases (Fig 3b), and otherwise better performance in SMARTS simulation
 - Interpretability of "most interactive participant" for each vehicle. Also the output \psi having the semantic meaning to help interpret the level of egotistical-vs-altruistic for each vehicle.

Weaknesses:
 - Over-complicated solution that is self-limiting. Why select only one other vehicle, to model interaction with, why not all, like [30] was already doing. Lines 154-155 describe limiting something that could have been more general, into something that is now more complicated *and* more limited in terms of only considering *one* other vehicle.
 - Unclear how it would actually work on a real robotics system. Difficult to understand this without a real platform or real data (e.g. Waymo or Argo open datasets). While the SMARTS benchmark tests high level coordination, the robots are treated as "self driven particles" (line 27), which makes me wonder how appropriate this work is for a *robotics* conference.
 - Initially difficult to understand the method: much abstract language initially discussing "social coordination", defining these terms unambiguiously too late. I think for a technical paper like this: get to the point faster. There's many terms like this including "social preferences" and "socially-compatible behaviors". The language of "dynamic social preferences" can mean many things, but seems a rather grandiose way to describe what's really happening, which is a weighted sum to two rewards that are only a function of the current observations (ignoring history).

**Summary Of Recommendation:**

I'd choose "borderline" if CoRL had such a recommendation, but without it: Weak Reject. See Weaknesses and Issues sections.

----

update: I thank the authors for their responses. Apologies I did not get to this sooner. But your responses have addressed most of my concerns. I have thus updated my score to accept

---

> ### Author Response · Authors · 2022-08-22
> **Response to the reviewer zrSF**
>
> We thank the reviewer for the insightful suggestions. We address the reviewer's comments and questions individually as follows, and we have also uploaded an revised version of our paper which now includes the supplementary material temporally. For the ease of reading, we have elaborated the details of the updates in our note to all reviewers. We hope that our response will enable the reviewer to appreciate the clarity of the paper and motivation as the other reviewers have.

---

> ### Author Response · Authors · 2022-08-22
> **Response to the reviewer**
>
> #### Comment 1:
> > "section 2: what about other works like Multiple Futures Prediction and Trajectron? Multi-agent autoregressive method with categorical latent variables can encode different types of behaviors or preferences too, and are (in addition) functions of the full history of past behaviors, while still reacting to latest observation (compared to this work which only looks at latest observation).".
> #### Answer:
> We greatly appreciate the reviewer’s effort in providing suggestions to improve the paper. In our revised paper, we have added the discussion of the several related works in Section 2.2. We think they are different from our paper because they are mainly designed for feature representation, which may be closely connected to the decision-making architectures like SAPO. However, they have inspired us to consider if we can make better use of the history of past behavior to achieve better performance. We have tried several ways for RL methods to consider the historical steps. In terms of the average success rate (i.e., the average ratio of the number of agents that reach their goals over the total number of agents in an episode.), we show the testing results of the trained model as follows. Note that each case are trained in 2e7 steps.
>
>
>
> Table 1. Ablations of considering history of past behaviors
>
> | Ablation \ Task                                 | Cross-road-v0 | Merge-v0  | Roundabout-v0 |
> |:----------------------------------------------- |:-------------:|:---------:|:-------------:|
> | Only use the latest observation (original SAPO) |   **0.923**   |   0.874   |     0.811     |
> | Use LSTM for memory                             |     0.799     |   0.821   |     0.672     |
> | Stack 3 steps of observations                   |     0.920     | **0.912** |     0.809     |
> | Stack 5 steps of observations                   |     0.917     |   0.889   |   **0.819**   |
> | Stack 10 steps of observations                  |     0.909     |   0.893   |     0.772     |
>
>
> From this table, we observe that **LSTM does not work very well in MASD systems**. The reason is that the coordination or conflicts of the AV's goals seldom keep the same in a whole episode, which make it more difficult to predict SPs based on the overall memory. On the other hand, we observe that **it may improve the performance of SAPO by stacking several steps of observations together**. This indicates that we should explore how to better use historical trajectory to predict SP and make decisions. Thus, we conclude the discussion as one of the limitations and future research directions (see Section 7 of the revised paper).
>
>
>
> #### Comment 2:
> > "equation 6c, 6d: should these be function of p?"
> #### Answer:
> In SAPO, policy and value network only need the representation vector from interactive Attention, without using the partner index $p$.
>
>
> #### Comment 3:
> > "line 171: "unreasonable behaviors such as sacrificing oneself to improve group reward", why is this bad? Isn't this just \phi=\pi/4 in equation 9? Especially when lines 245-246 say that summed group reward is the evaluation criteria anyway."
> #### Answer:
> We thank the reviewer for this comment. Considering each AV are in independent control, it is not good to use $\phi=\pi/4$ bacause AV will easily miss its goals and cannot figure out what is its own target destination and what is the other's. Our ablation studies have shown that using fixed SPs like $\phi=\pi/4$ will need more time to improve the policies (see Figure 5(b) of our paper). Actually, for the evaluation metric, we utilize the total of individual reward because SMARTS always set great penalties to an AV if it does not reach the goal at the end of the episode or collide with other vehicles. Thus, we can see some gaps if we evaluate the total reward of effcient traffic navigation and congestions in MASD systems. Note that we have used the average success rate as a more common metric, instead of reward (according to Reviewer JLzw's advice).
>
>
> #### Comment 4:
> > "section 5.1: while SAPO trains 5x faster than IPPO, how do we know this isn't just poor hyperparameter choice for IPPO? It asymtotes the same. Also, what are the uncertainty regions on Figure 3, what do they represent?"
> #### Answer:
> For PPO part, we use the same hyperparameters (see detailed hyperparameter settings in Section E in Appendix). We use the uncertainty regions because we evaluate the trained policies in multiple random seeds and get the maximum, mean and minimum results to visualize the results. We have charified the representation in Section 5 of our revised paper.
>
> #### Comment 5:
> > "lines 183-184: mention "corresponding to the latest traffic conditions" but this seems to be overselling: it's only a function of the current observations. There is no history..."
> #### Answer:
> We thank the reviewer for this comment. We have clarified this sentence in our revised version.

---

> > ### Author Response · Authors · 2022-08-22
> > **Response to Reviewer zrSF continued...**
> >
> > #### Comment 6:
> > > "line 211: "(a) removing Interactive Attention": a more interesting ablation here would have been random Interactive Attention. This would isolate-away the selection of p, and test how good the socially-aware part of the method would be. Possible an indication of how "toy" these experiments are would be: is the performance just the same? I imagine it could be, but for any sufficiently complicated setup (perhaps what is shown in figure 6, or some more complicated traffic simulator) the ablation would show just how necessary it is to selection p intelligently."
> > #### Answer:
> > We greatly appreciate the reviewer’s effort in providing suggestions to improve the paper. In the ablation studies of our revised paper, we have added more ablation studies about how to choose the interacting partner, by using other mechanisms to select $p$ (i.e., to use random attention or a rule of selecting the nearest vehicle). From the results of Figure 5(a), we show that random attention cannot perform well, while using some greedy rules can improve the performance of SAPO directly, but still need improvements to achieve state-of-the-art performance. That is, Interactive Attention can provice an promising direction of choosing the interacting partner, before we have got a series of well-defined rules. Besides, in Section 5.2.1, we also give a detailed description of how to make decisions with SPs without using Interactive Attention.
> >
> > #### Comment 7:
> > > "Why does each vehicle only select the most interactive other vehicle? there's no reason it can't focus on all vehicles, similar to [30], which inspired this work?"
> > #### Answer:
> > To our perspective, Social Attention [30] can consider all of vehicles with different attention weights, based on the centralized control of multiple AVs. To remove the impacts of the control method, we also consider the variant of decentralized control, where each AV maintains its own attention and PPO framework independently. We have compared with these two versions of Social Attention with SAPO in Figure 7. We observe that Social Attention can consider all of vehicles but it will also keep much redundent information in the features, which will slow down or do harm to the training process. Besides, as shown in Figure 5(a) from the ablation studies, there also exist the similar conslusion about the importance of choosing partners intelligently.
> >
> >
> > #### Comment 8:
> > > "Clarity: I think reading the abstract and introduction, it was difficult to understand what was actually being proposed. There were mentions of abstract benefits like "social preferences", "social-aware integration mechanism to integrate objectives of interacting AVs", and even "dynamic social preferences", and "socially-compatible behaviors", but it'd be good to get to the point faster. At line 54 it's still unclear what "socially-compatible behaviors" means, also on line 73 and 76, would help clarify to define this concretely earlier."
> > #### Answer:
> > We greatly appreciate the reviewer’s effort in providing suggestions to improve the paper. To get to the point faster, we have clarified the abstract and introduction in our revised paper. For example, we remove the mention and description of several abstract words, such as "self-particle", "social-compatible behaviors" and "dynamic SPs".
> >
> >
> > #### Comment 9:
> > > "line 63: MARL is widely used to navigate AVs in MASD systems" perhaps clarify this means the research litreaeture, not real AVs? Hardly any real AVs use RL, let alone MARL."
> > #### Answer:
> > We thank the reviewer for this comment. We have clarified this description in our revised paper.
> >
> >
> >
> > #### Comment 10:
> > > "equation 3: doesn't the advantage function usually (1) not take a reward as input and (2) is a function of the expected reward as Q-V"
> > #### Answer:
> > We thank the reviewer for this comment. For issue (1), we have revised the left term of the Equation 3, since the reward is usually a function about state and action. For issue (2), we should clarfiy that the advantage function has followed the common defintion in PPO, denoted by: $A(o_t,a_t)=r(o_t,a_t)+\gamma\cdot V(o_{t+1})-V(o_t)$, where $r(o_t,a_t)+\gamma\cdot V(o_{t+1})$ can be seen as an unbised estimation of $Q(o_t,a_t)$ and we do not need to predict $Q$ by an extra neural network in PPO. In our real implementation, we use Generalized Advantage Estimation (GAE) to eatimate a more instructive value function.

---

> > > ### Author Response · Authors · 2022-08-22
> > > **Response to Reviewer zrSF continued...**
> > >
> > > #### Comment 11:
> > > > Figure 2 (yellow box): it is shown how r^SP: goes between egotistic (\phi=0) and alteristic (\phi=\pi/2). But equation 8 mentions the range can go to -pi/2 radians too. Is this intended? I.e. ego acting "maliciously", against the other vehicles interests? If so, should it be shown in Figure 2 as well, or should the range of \phi in equation 8 be limited to [0, \pi/2]?
> > > #### Answer:
> > > We thank the reviewer for this comment. In our revised paper, we have modified our Equation 8 for consistency. We mainly consider $[0,\pi/2]$ in all of MASD systems supported by SMARTS, including merging or passing in multi-lane traffic. This is because the SP in $[-\pi/2,0]$ may bring some aggressive or unsafe behaviours. However, Schwarting et al. also mentioned that $\phi\in[-\pi/2,0]$ may also be useful to form some competitive behaviours in car racing scenarios.
> > >
> > >
> > > #### Comment 12:
> > > > lines 174-176: different how? perhaps \emph{} which specific part following "[20]" that is different?
> > > #### Answer:
> > > We thank the reviewer for this comment. We have revised the sentences in Section 4.2, by adding more descriptions of CoPO[20]. Besides, we also add the comparison with Peng's CoPO in SMARTS, and charify the main differences between CoPO and SAPO in Section 5.2.3, and here we will explain them in details:
> > > 1. In terms of choosing the interacting partners, CoPO focus on all of the surrounding vehicles within "a fixed range" (need to be tuned carefully as an extra hyperparameter), while SAPO focus on one of the surrounding vehicles by using Interactive Attention. We have discussed the advantages of Interactive Attention in our ablation studies and comparison with Social Attention (decentralized).
> > > 2. CoPO fixes the SP of each AV at each timestep and adjusts it episodically by using the total reward of all active agents in Metadrive. Differently, SAPO updates the SP step by step, by only utilizing the current observation of the ego AV. However, considering multiple training cases together, we think the coordination or conflicts of the AVs' goals will not keep the same in a whole episode. This is the main reason why CoPO cannot perform very well in some scenarios (see Figure 7).
> > > 3. From the previous points of differences, it can be concluded that CoPO is mainly designed for centrailized training and decentralized execution (CTDE) of multiple AVs. For MASD systems, it is an interesting and powerful framework in "multi-agent control", where AVs should be trained together. However, without the dependence of the global information, SAPO can not only work in "multi-agent control" (by training multiple AVs independently), but also in "single-agent control" (by training a single AV to interact with social vehicles). We have discussed how to apply SAPO in a single-agent self-driving environment (see Section D of the Appendix).
> > >
> > >
> > > #### Comment 13:
> > > > lines 197: "goals in orders": which order? does one particular car need to go first?
> > > #### Answer:
> > > We thank the reviewer for this comment. We charify the explanation in our revised version, where AVs have learned spontaneous orders to pass the intersection, navigated by its well-trained policy. Note that we do not define which particular car need to go first.
> > >
> > >
> > > #### Comment 14:
> > > > line 231: who were vehicles AV-0 and AV-3 each selecting, was it respectively AV-2 and AV-1?
> > > #### Answer:
> > > During the stage(a), AV-0 and AV-3 selects AV-2 or AV-1. We use hard attention to overcome the useless information in the AV's observation. Thus, they only focus on one of them at each timestep.
> > >
> > >
> > > #### Comment 15:
> > > > Other small points, including:
> > > > line 23 ("inreasing" --> "increasing")
> > > > line 72 ("response" --> "responses")
> > > > line 95 ("the policy space \mathcal{U}" --> " \Pi")
> > > > equation 6a: "p \in \mathcal{M}" --> "p \in \mathcal{M} \ {m}"?
> > > > line 159: "features" --> "generalization"
> > > > figure 6: maybe use white background, and zoom in more.
> > > #### Answer:
> > > We thank the reviewer for this comment. In the revised version, we have fixed all of these typos, syntax and errors to improve the presentation quality.

---

### Official Review · Reviewer_U4bM · 2022-08-10

**Originality:** Very Good
**Technical Quality:** Excellent
**Clarity Of Presentation:** Excellent
**Impact:** 3

**Recommendation:**

Strong Accept: I recommend accepting the paper and will argue for my recommendation even if other reviewers hold a different opinion.

**Summary:**


This work proposes a MARL method called Socially-Attentive Policy Optimization (SAPO) for controlling autonomous vehicles to behave socially-compatible behaviors.

Two key components are used:

1. Self-attention module foot select most interactive participant for each AV
2. Social-aware reward shaping mechanism.

SAPO enables varying the social preferences of an agent within one episode. Evaluation results show that SAPO can improve training efficiency and also emerge socially-compliant behaviors.

**Issues:**


In related work [Peng et al.], the authors use a driving simulator MetaDrive for investigating the detailed behaviors in traffic flow. This work use SMARTS benchmark where only limited number of vehicles coexists. Also, I would expect to see the comparison with the algorithm CoPO proposed in [Peng et. al.], since both use SVO measurement to fuse reward. For the completeness of this work, I would suggest to run CoPO in SMARTS or run SAPO in MetaDrive to fairly compare related methods.


[Peng et al.] Learning to Simulate Self-driven Particles System with Coordinated Policy Optimization


---


I am a little worried about Figure 6. We see SP maintains fixed in ~75 degree and ~10 degree when AV-3. Are there exist hard limit of SP? How can the neural network predicted SP be so smooth within one episode?


**Quality Of The Limitations Section:**

Limitations are addressed clearly

**Reviewer Expertise:**

5: The reviewer is absolutely certain that the evaluation is correct and very familiar with the relevant literature

**Robotics Focus:**

Highly relevant to robotics but no hardware experiments

**Strengths And Weaknesses:**


### Strengths

* The method is concise and sound.
* The presentation is clear and smooth.

### Weaknesses

Generally, I can't find any major weakness that harms this work.

* Lack of baselines. (See Issues)
* Figure 2 can be simplified a little bit.



**Summary Of Recommendation:**


Generally, I like this work. The work addresses the social behaviors in traffic flow by first discovering the issues in current algorithms, namely the dynamic social preferences within one episode, and then proposing solutions accordingly.
The presentation is clear and the method is sound and reasonable. The evaluation on SMARTS simulator shows promising results.

I have two concerns. First is the lack of baselines (see Issues). The second is whether this work should be presented in CoRL since there are no real robot experiments nor discussion on the applicability of SAPO in real-world. Nevertheless, I admit that the social behaviors in multi-agent systems is indeed an important topic, since in future we may co-live with those robots.

---

> ### Author Response · Authors · 2022-08-22
> **Response to the reviewer U4bM**
>
> We thank the reviewer for the insightful suggestions. We address the reviewer's comments and questions individually as follows, and we have also uploaded an revised version of our paper which now includes the supplementary material temporally. For the ease of reading, we have elaborated the details of the updates in our note to all reviewers.

---

> ### Author Response · Authors · 2022-08-22
> **Response to the reviewer**
>
> #### Comment 1:
> > "the lack of baselines".
> #### Answer:
> We thank the reviewer for this comment. We have added the comparison with CoPO in SMARTS environment. The comparison results are shown in Figure 7 of our revised paper. We also charify the main differences between CoPO and SAPO in Section 5.2.3, and here we will explain them in details:
> 1. In terms of choosing the interacting partners, CoPO focus on all of the surrounding vehicles within "a fixed range" (need to be tuned carefully as an extra hyperparameter), while SAPO focus on one of the surrounding vehicles by using Interactive Attention. We have discussed the advantages of Interactive Attention in our ablation studies and comparison with Social Attention (decentralized).
> 2. CoPO fixes the SP of each AV at each timestep and adjusts it episodically by using the total reward of all active agents in Metadrive. Differently, SAPO updates the SP step by step, by only utilizing the current observation of the ego AV. However, considering multiple training cases together, we think the coordination or conflicts of the AVs' goals will not keep the same in a whole episode. This is the main reason why CoPO cannot perform very well in some scenarios (see Figure 7).
> 3. From the previous points of differences, it can be concluded that CoPO is mainly designed for centrailized training and decentralized execution (CTDE) of multiple AVs. For MASD systems, it is an interesting and powerful framework in "multi-agent control", where AVs should be trained together. However, without the dependence of the global information, SAPO can not only work in "multi-agent control" (by training multiple AVs independently), but also in "single-agent control" (by training a single AV to interact with social vehicles). We have discussed how to apply SAPO in a single-agent self-driving environment (see Section D of the Appendix).
>
>
> #### Comment 2:
> > "whether this work should be presented in CoRL since there are no real robot experiments nor discussion on the applicability of SAPO in real-world."
> #### Answer:
> We greatly appreciate the reviewer’s effort in providing suggestions to improve the paper. This paper mainly evaluates the algorithm in SMARTS, a well-designed simulation platform which mainly focus on the interactive behaviours of AVs. To further address the applicability of SAPO in real-world, we have revised our paper in the following three aspects:
>
> 1. **Evaluation metric:** According to reviewer JLzw's advice, we use a more common metric called "average success rate" to evaluate performance of the trained models in all of our self-driving experiments, instead of the abstract reward given by SMARTS. We calculate this metric by using the average ratio of the number of agents that reach their goals over the total number of agents in one testing episode.
> 2. **Randomness:** According to reviewer 6Syb's advice, we give more implementation details about the randomness of the training cases in SMARTS and how to deal with it by using the distributed RL architecture like RLlib (see Section A and B in the Appendix). However, it is still not enough to demonstrate that the SAPO agents is well-designed to deal with a diverse set of other agents. Thus, we enrich the limitation section of this paper, by considering this will be a promising start point of apply RL in real-world applications.
> 3. **Real application:** According to Area Chair Yub7's advice, we evaluate SAPO in a more realistic self-driving system, where SAPO controls a single AV to interact with the real traffic based on Waymo Motion Dataset and Argoverse Dataset. Considering SAPO works as an independent controller of each AV and does not require any global information for training, we can directly test SAPO in this single-agent control scenario, without major modifications. In Appendix, we have shown some potential applications. However, we also find some open questions of replaying the RL-based vehicles and data-driven vehicles together in an environment. More details are given in Section D of the Appendix.
>
>
>
> #### Comment 3:
> > "I am a little worried about Figure 6. We see SP maintains fixed in ~75 degree and ~10 degree when AV-3. Are there exist hard limit of SP? How can the neural network predicted SP be so smooth within one episode?"
> #### Answer:
> Apologize not to introduce clearly. To predict SP more smoothly, we discretize $\varphi$ at each testing timestep, as $\varphi\in[0,15,30,45,60,75,90]$. As the output of the neural netwoek, we sample $\varphi$ from a categorical distribution, instead of directly outputing a continuous value from a gaussian distribution. Besides, as AV-3 yields the other vehicles at the beginning (see the bottom of Figure 6), the feature representation of its obserevation does not change sharply, resulting in the smooth output of SP.
>
>
> #### Comment 4:
> > Figure 2 can be simplified a little bit.
> #### Answer:
> We thank the reviewer for this comment. We have simplified some elements of Figure 2.

---

> > ### Comment · Reviewer_U4bM · 2022-08-25
> > **Thanks for revision and response**
> >
> > Most of my issues are addressed in rebuttal. One more question here is that why new CoPO results suggest that CoPO is so stable even compared to independent PPO? In my understanding CoPO should at least perform as good as PPO since it only need to set SVO to 0 degree and the method reduces to IPPO.

---

> > > ### Author Response · Authors · 2022-08-25
> > > **Response to the reviewer**
> > >
> > > In our opinion, the main reason is that CoPO utilizes a meta learning technique. Actually, CoPO and SAPO both utilize SVO to integrate the individual rewards of interactive agents together as the learning objective, with PPO as the start point of design. They learn how to adjust SVO by deep learning methods instead of
> > > artificial settings (e.g., set SVO to a fixed degree directly). The difference is that CoPO is a 2-step optimization method, which considers: (1) to maximize the social-aware integration of the individual reward and neighborhood reward for learning PPO policy; and (b) maximize the global reward of all active agents for updating SVO. As a result, CoPO is more dependent on the fune-tuned reward to get reasonable SVO and stable policies, compared to SAPO and IPPO which are both 1-step optimization methods and do not need global reward. That is, if optimizing the global reward is very unstable for policy gradient methods in some difficult scenarios, we may observe that CoPO is even more unstable than IPPO.
> > >
> > > Just as in many works considering the meta-learning process like CoPO, we observe that the authors utilize several useful techniques in their official code to improve the final performance (see: https://github.com/decisionforce/CoPO/blob/main/copo_code/copo/train_all_copo_dist.py), such as population based training (PBT) to search the best start seed and the range of neighbors for each AV. Based on RLlib, we think it is a high quality code. We implement CoPO by migrating this code from Metadrive to SMARTS, since both IPPO and SAPO are based on RLlib as well.

---

> > > > ### Comment · Reviewer_U4bM · 2022-08-25
> > > > **Thanks for response**
> > > >
> > > > I think the rationale is convincing. I will raise my score based on the revision and the responses.
> > > >
> > > > (Opps I can't find the bottom to edit my original review, maybe I don't have the right to change my score?)

---

> > > > ### Comment · Reviewer_U4bM · 2022-08-25
> > > > **By the way**
> > > >
> > > > BTW, I think there is no "PBT to search best start seed" for CoPO. The code you are referring shows that they use ray.tune to help running multiple trials with different start seeds for repeated experiments.

---

### Author Response · Authors · 2022-08-22
**Note to all Reviewers**

We would like to thank the reviewers for their careful reading and insights that have improved the paper with more comprehensive analyses. Additionally, we thank the reviewers who have expressed their appreciation of the importance of the problem we tackle and the clarity of our exposition.

The purpose of the paper is to introduce a framework within reinforcement learning that easily incorporates existing methods enabling them to efficiently solve problems in which the controller faces social preferences for taking actions. Our experiments verify our claim that such technology improves RL performance and enables RL methods to consistently solve problems of this kind where existing methods perform badly and may sometimes even fail.


**We outline some of the items we have now added to the manuscript to complement the already in-depth analyses we have performed in the paper:**
* We add more baselines in Section 5.2.3, including: (a) another famous MASD method considering social preference, called CoPO [Peng et al.]; and (b) the variant of Social Attention using decentralized control like SAPO.
* We discuss some multi-agent autoregressive methods in Section 2.2. They are mainly designed for trajectory prediction, which are not the baselines of decision-making scenarios. However, on the other hand, we observe that all of them make good use of the history, which inspire us to add an extra experiment to explore whether SAPO can get improved by using LSTM or stacking sequences of observations in multiple steps (see answers to Reviewer zrSF).
* We add more ablation studies about how to choose the interacting partner intelligently in Figure 5(a). Here we clarify why the ego-vehicle should not focus on all of the surrounding vehicles.
* We evaluate SAPO in two more realistic self-driving systems, where SAPO controls a single AV to interact with the real traffic based on Waymo Motion Dataset and Argoverse Dataset (see Section D in Appendix).
* We utilize a common metric called "average success rate" to evaluate the performance of algorithms in SMARTS experiments, instead of using the abstract reward given by SMARTS.
* We clarify the abstract and introduction to make readers get the clear key point. Besides, we highlight the differences between SAPO and our baseline CoPO in Section 5.2.3. As SAPO is designed to train an AV independently without global information, we clarify that SAPO can also be suitable in a single-agent control system where one AV is navigated by SAPO and the others are navigate by real dataset.
* We fix all of the writing issues from reviwers' advice and improve the usage of the vocabulary. We give early defination of the algorithm details in our revised paper.
* We have improved the limitation sections by addressing all reviewers' concerns.


We hope the reviewers appreciate the updates to our paper.

---

### Comment · Area_Chair_Yub7 · 2022-08-25
**Engagement with authors regarding their responses to the reviews**

Dear reviewers, the authors have responded in detail to the reviews.  It would be great if you could further engage with the authors now, as the author/reviewer discussion/rebuttal phase ends Aug 27 at 11:59 PM Pacific. Many thanks in advance -- your participation greatly contributes to the overall quality and value of the review process!
-- your Area Chair

---

### Meta-Review · Area_Chair_Yub7 · 2022-08-14

**Recommendation:** Accept (Poster)
**Confidence:** 5

**Metareview:**

Updated Meta Review

We now have:  strong accept (x1), weak accept (x3), so the discussion has mostly converged.
The paper has seen significant updates in response to the reviewers, and upgraded recommendations from two of the four reviewers.
The remaining weakness is the first one listed below, namely that the test scenarios still use either non-reactive vehicle trajectory data, or co-trained agents.  So working with a diverse and realistic set of driving behaviors is left as future work.

Overall,  the reviews point to the paper making a worthy contribution, even with the above remaining weakness. My recommendation is therefore to Accept (Poster).

Strengths:
- important topic
- original ideas & sound method:
  introduces self-attention to select most interactive "other AV" for each AV in scene;
  introduces social awareness, by estimating the dynamic social preferences from their observations.
- well written
- up to 5x faster learning for some cases (Fig 3b) compared to several baselines, for MASD scenarios in SMARTS

Weaknesses:
- the motivation (better interaction with all agents)
  is ultimately not well matched with the method, where all agents are trained together; even with the additional experiments, there is no hard evidence that the algorithm can train a driving agent to interact with other human-driver responses to the ego-vehicles actions, or at least agents not co-trained with it.
- lack of baselines, e.g., other multi-agent autoregressive methods that have latent variables that
  could also encode preferences;    (now addressed with further baselines)
- in some scenarios, the ego-vehicle should take into account more than one agent. (partly addressed via additional ablations)
- no real robot experiments, nor discussion on the applicability of the algorithm & results in the real world.  SMARTS tests are distant from real-world scenarios, could have used Waymo or Argo open datasets.   (now evaluated with these more realistic datasets with the notable caveat that the agents in these datasets are not reacting to the ego-agents actions;  should be clarified whether the results come from training or validation parts of the dataset)
- somewhat obfuscated vocabulary and missing early definitions (now improved)
- limitations section needs improvement (now improved)

---

> ### Author Response · Authors · 2022-08-22
> **Response to the Area Chair Yub7**
>
> We thank the reviewer for the insightful suggestions. We address the reviewer's comments and questions individually, and we have also uploaded an revised version of our paper which now includes the supplementary material temporally. For the ease of reading, we have elaborated the details of the updates in our note to all reviewers. We hope that our response will enable the reviewer to appreciate the clarity of the paper.

---

> ### Author Response · Authors · 2022-08-22
> **Response to Area Chair**
>
> #### Comment 1:
> > "lack of baselines, e.g., other multi-agent autoregressive methods that have latent variables that could also encode preferences;".
> #### Answer:
> We add more baselines in Section 5.2.3, including: (a) another famous MASD method considering social preference, called CoPO [Peng et al.]; and (b) the variant of Social Attention using decentralized control like SAPO. Besides, we discuss some multi-agent autoregressive methods in Section 2.2. They are mainly designed for trajectory prediction, which are not the baselines of decision-making scenarios. However, on the other hand, we observe that all of them make good use of the history, which inspire us to add an extra experiment to explore whether SAPO can get improved by using LSTM or stacking sequences of observations in multiple steps (see answers to Reviewer zrSF).
>
>
> #### Comment 2:
> > "in some scenarios, the ego-vehicle should take into account more than one agent.".
> #### Answer:
> We add more ablation studies about how to choose the interacting partner intelligently in Figure 5(a). Here we clarify why the ego-vehicle should not focus on all of the surrounding vehicles.
>
> #### Comment 3:
> > "no real robot experiments, nor discussion on the applicability of the algorithm & results in the real world. SMARTS tests are distant from real-world scenarios, could have used Waymo or Argo open datasets".
> #### Answer:
> We evaluate SAPO in two more realistic self-driving systems, where SAPO controls a single AV to interact with the real traffic based on Waymo Motion Dataset and Argoverse Dataset (see Section D in Appendix). For more physical meanings, we utilize a common metric called "average success rate" to evaluate the performance of algorithms in SMARTS experiments, instead of using the abstract reward given by SMARTS.
>
>
> #### Comment 4:
> > "the motivation (better interaction with all agents) is ultimately not well matched with the method, where all agents are trained together".
> #### Answer:
> We clarify the abstract and introduction to make readers get the clear key point. Besides, we highlight the differences between SAPO and our baseline CoPO in Section 5.2.3. As SAPO is designed to train an AV independently without global information, we clarify that SAPO can also be suitable in a single-agent control system where one AV is navigated by SAPO and the others are navigate by real dataset.
>
>
> #### Comment 5:
> > "somewhat obfuscated vocabulary and missing early definitions".
> #### Answer:
> We fix all of the writing issues from reviwers' advice and improve the usage of the vocabulary. We give early defination of the algorithm details in our revised paper.
>
> #### Comment 6:
> > "limitations section needs improvement".
> #### Answer:
> We have improved the limitation sections by addressing all reviewers' concerns.